# Investigation of Strain Fatigue Behavior for Inconel 625 with Laser Shock Peening

**DOI:** 10.3390/ma15207269

**Published:** 2022-10-18

**Authors:** Yaofei Sun, Han Wu, Haifeng Du, Zhenqiang Yao

**Affiliations:** 1School of Mechanical Engineering, Shanghai Jiao Tong University, Shanghai 200240, China; 2State Key Laboratory of Mechanical System and Vibration, Shanghai 200240, China

**Keywords:** strain fatigue, Inconel 625, laser shock peening, dislocations, twins, crack initiations, fatigue striations

## Abstract

With excellent creep resistance, high-temperature thermal strength and high-temperature fatigue strength, Inconel 625 is widely applied to fabricate structural components in the aerospace field, where fatigue life is a key point. Laser shock peening (LSP) is considered to improve the fatigue strength and fatigue crack growth resistance of metal materials. The present work was conducted to investigate the influence of LSP on strain-controlled fatigue behavior of Inconel 625. The surface microstructures of specimens before and after LSP were observed by transmission electron microscope (TEM). The strain-controlled fatigue loading tests with different strain amplitudes ranging from 0.4% to 1.2% were carried out on the specimens, and the topography of fracture appearance was examined by scanning electron microscope (SEM). The investigations showed that the specimens with LSP presented fewer crack initiations, shorter fatigue striations space and smaller dimples or micropores, which account for the enhancement of the fatigue life for the LSP specimens. Furthermore, the plastic deformation, ultra-fine grains, twins and dislocations caused by LSP could prevent crack initiation, crack propagation and ultimate fracture, hence prolonging the fatigue life of the Inconel 625. In addition, it was revealed that the cyclic strain hardening as well as cyclic strain softening remains almost the same to Inconel 625 with or without LSP.

## 1. Introduction

Nickel-based superalloy with nickel as the matrix, adding W, Mo, Ti, Al, Nb, Co and other strengthening elements, has excellent high-temperature thermal strength, creep resistance and high-temperature fatigue strength [1]. The material is widely applied to fabricate structural components [2] in the aerospace field of 900~1050 °C temperature, such as aero-engine blades [3,4,5], turbine discs [6], combustors [7], etc. Its excellent chemical and metallurgical properties are mainly reflected in the nickel base alloy being able to dissolve a variety of alloying elements and maintain good structural stability. Inconel 625 alloy is a nickel-based deformed superalloy containing a large amount of Cr, Mo and Fe elements and Nb as the main additive elements.

A lot of research has been carried out on the fatigue characteristics of metal materials including nickel alloys. Wright et al. [8] investigated the low-cycle fatigue (LCF) behavior of Inconel 617 at 850 °C and 950 °C. Rao et al. [1] determined the influence of the temperature on the low-cycle fatigue for Inconel 617 alloy at room temperature and 750 °C. Zhou et al. [9] carried out full strain controlled low-cycle fatigue tests with strain amplitudes ranging from 0.3% to 1.5% for two 316 L austenitic stainless steels. The results show that both steels undergo initial hardening followed by a longer range of softening, especially at lower strain amplitudes (0.3% to 0.5%). The cyclic deformation behavior of Fe–Ni–Cr alloy with different strain amplitudes was investigated by a low-cycle fatigue (LCF) test at 700 °C [10]. It was found that Sanicro 25 alloy has cyclic hardening behavior. Li et al. [11] compared cyclic deformation behavior of all-austenitic stainless steel 316 L and duplex stainless steel 2205 with different strain amplitudes, loading rates and tensile holding times. The results show that the cyclic stress response of 2205 is primary cyclic hardening followed by long-term cyclic softening, while 316 L shows secondary hardening at larger strain amplitudes. Xing et al. [12] investigated the fatigue properties of Inconel 690. The results of cyclic stress–strain response investigation showed that the response strongly depended on strain amplitude.

In laser shock peening (LSP), a laser beam with a duration in the nanosecond range and an energy intensity in the gigawatt scale [13,14] impinges the sacrificial coating on the surface of the metallic target and forms plasma, which expands rapidly, generating a strong shock wave into the bulk [15]. These shock waves induce plastic deformation on the metal surface, resulting in compression residual stresses [16]. LSP generates microstructure modification, including grain size reduction [17,18], dislocation proliferation and rearrangement [19,20,21] and mechanical twin generation [22]. Górnikowska et al. [23] investigated the influence of LSP on the topography, microstructure, surface roughness and the mechanical properties of the Inconel 625 nickel alloy. It was concluded that the LSP caused severe plastic deformation of the surface layer, resulting in the slip bands on the surface and cross-section of the treated material, a high density of dislocations and a higher hardness of the treated region. Su et al. [24] applied LSP to AA6061-T6 welded joints and confirmed by experimental results and numerical analysis that the thermal corrosion resistance and fatigue properties of the welded joints were improved. Spadaro et al. [25] studied the effects of laser uncoated impact strengthening and shot peening (SP) on the low-cycle fatigue life of 253 MA austenitic stainless steel. The results showed that the stability of near-surface quenched microstructure improved the fatigue life of shot peened 253 MA. Sun et al. [26] investigated the difference of different laser shock strengthening zones on high-cycle fatigue properties of 42CrMo high-strength steel. The results showed that surface and side shot peening had better fatigue life than only surface shot peening.

Much research had been carried out on the low-cycle fatigue of Inconel 718 and other alloys. The effects of LSP on the surface integrity, microstructure, strength, hardness and fatigue properties of materials had been researched. However, very few investigators focused on the strain fatigue behavior of Inconel 625, let alone Inconel 625 after LSP. In addition, the correlation between the surface modification caused by LSP and the fatigue strength of Inconel 625 needs further research. Therefore, in this paper, low-cycle fatigue tests of an Inconel 625 specimen before and after LSP were carried out. The point of this paper was to understand the mechanism of fatigue life improvement by LSP. The strengthening effect of LSP on Inconel 625 was revealed by transmission electron microscope (TFM) observation, and the fracture topography of specimens with and without LSP was investigated by scanning electron microscope (SEM).

## 2. Experiments Setup

### 2.1. Specimen Preparation of Inconel 625

The chemical components of Inconel 625 are exhibited in Table 1. To study the effect of LSP on fatigue performance, the Inconel 625 plate with length of 300 mm, width of 150 mm and thickness of 8 mm was processed with laser shock. During the process, the surface of the sample was covered with uniform black tape as the absorption layer and water as the constraint layer. The sample is fixed on the clamp at the end of the robot, and the robot drives the sample in accordance with the prescribed trajectory, which is calculated by MATLAB simulation. In the LSP, the Nd:YAG pulsed laser is used, whose parameters are shown in Table 2.

According to the Chinese National Standard (GB/T 26077-2021), the Inconel 625 without LSP and with LSP were processed into the shape and size shown in Figure 1 using wire-cut electrical discharge machine (LT400, POSITTEC, Suzhou, China). The length and thickness of the sample are 141 mm and 8 mm. The width of the clamping end and the test section are 24 mm and 12 mm. Then, the surface of the specimen was polished by automatic polishing machine (MECATECH 300 SPS, PRESI, Grenoble, France) and manually polished to ensure the roughness of the fatigue test range area: Ra < 0.32 μm.

### 2.2. Fatigue Test Machine and Parameters

As demonstrated in Figure 2, strain fatigue tests were carried out in air at room temperature on MTS Landmark electro-hydraulic servo fatigue test machine manufactured by MTS Metes Industrial Systems Co., Ltd, Eden Prairie, MN, USA. The testing machine has the following main parameters: dynamic load of ±500 kN; strain loading speed ranging from 0.05%/s to 1%/s; displacement of ±90 mm; resolution of 0.0001 mm; fully digital closed-loop control; and the control system can automatically calibrate the testing machine accuracy (load, strain, displacement) and then realize automatic adjustment.

According to the national standard (GB/T 26077-2021), a symmetrical triangular wave shape R = −1 was applied to the specimens during the strain-controlled fatigue loading tests. In addition, five strain amplitudes were selected for the test, which were 1.2%, 1.0%, 0.8%, 0.6% and 0.4%, respectively. The strain loading rate of each group was 0.4%/s, which was measured by an extensometer installed in the test sections of specimens. At such large strain amplitude, low-cycle fatigue fracture occurred.

### 2.3. Microscopic Characterization

TEM (Talos F200X, FEI, Hillsboro, OR, USA) operating at a voltage of 200 kV was utilized to observe the microstructures of specimens with and without LSP before fatigue loading test. Samples of TEM observation were sectioned perpendicular to specimen axial direction using ultra high-resolution scanning electron microscope (GAIA3, TESCA, Brno, Czech Republic) with Ga ion beam, and the thickness of samples was 100 nm. The topography of the specimens before and after LSP fracture surface was examined by scanning electron microscopy (VEGA 3, TESCAN, Brno, Czech Republic). Moreover, microhardness tests were conducted on the specimens’ surface using microhardness tester (HXD-1000TMC/LCD, Baoleng, Shanghai, China). A load of 5 N with a duration of 15 s was applied for each indentation.

## 3. Results and Discussion

### 3.1. Strain Fatigue Life

Strain loading test on the specimen was carried out once at each strain amplitude. The strain fatigue life of Inconel 625 before and after LSP is illustrated in Figure 3. As the strain amplitude increased from 0.4% to 1.2%, the fatigue life of specimens with LSP decreased from 9300 cycles to 562 cycles and the fatigue life of specimens without LSP decreased from 19460 cycles to 720 cycles, which both belonged to the low-cycle fatigue region. Colin and Fatemi et al. [27] investigated the fatigue behaviors of stainless steel 304 L in strain-controlled fatigue loading tests. The results showed the negative correlation between the fatigue life of stainless steel 304 L and strain amplitude. At the lowest strain amplitude (strain amplitude of 0.4%), the life of specimens after LSP, which exceeds 10^4^ cycles, is obviously higher than that of specimens without LSP. At a higher strain range (0.6–1.2%), the fatigue life of the specimen after LSP is close to that of the specimen without LSP but slightly higher than that of the latter.

### 3.2. Cyclic Strain Hardening and Softening Behavior

From the evolution of cyclic stress for Inconel 625 at different strain amplitudes as shown in Figure 4, the specimens before and after LSP exhibit cyclic strain hardening behavior prior to cyclic strain softening behavior until fracture. León-Cázares et al. [28] found that Inconel 718 alloy also showed the same cyclic stress–strain behavior through fatigue loading tests. Furthermore, the stress level increased with the increase of the strain amplitude for reinforced specimens and unreinforced specimens. The positive dependence of the maximum stress responses on strain amplitude and the negative dependence of the number of cycles of the maximum stress responses occurring on the strain amplitude are demonstrated in Figure 5 and Figure 6, respectively. Ilya et al. [29] discovered that cyclic stress of Fe–15Mn–10Cr–8Ni–4Si austenitic stainless steel was positively correlated with the strain amplitude. Moreover, at five different strain amplitudes, the maximum cyclic stresses and the cycle numbers corresponding to the maximum stress of reinforced specimens and non-reinforced specimens are close to each other. With comprehensive analysis of Figure 4, Figure 5 and Figure 6, the conclusion that the cycle number corresponding to the maximum stress decreases with the increase of the maximum stress can be drawn.

### 3.3. Microstructures and Hardness

The microstructures of the specimens before and after LSP were investigated by transmission electron microscope (TFM). The results are shown in Figure 7. For the specimen with LSP, at the depth of 1 μm, the grain size is smaller than 50 nm, i.e., the grain in this region is an ultra-fine grain; at the depth of 2 μm, there is an obvious boundary between ultra-fine grain and fine grain; at the depth of 5 μm, coarse grain could be observed whose size exceeds 1 μm. In addition, the micrograph at the depth of 6 μm exhibits geometric characteristics of the matrix phase. Lan et al. [30] found a similar gradient grain structure in a nickel-based alloy after LSP. For specimens without LSP, the grain size changes little along the depth direction. Furthermore, the grain size of the specimen without LSP is similar to that of the specimen with LSP at the depth 3 μm ~6 μm, which indicates that LSP only strengthens the layer with thickness of 1–3 μm from the surface. Figure 8 presents the twins and dislocations in the subsurface (about 1μm deep from the surface) of the LSP sample, which cannot be observed in the normal sample. After LSP, twins and high-density dislocations were also discovered by Telang et al. [31] in Inconel 718.

The microhardness of specimens without LSP and with LSP is demonstrated in Figure 9. The microhardness values of the Inconel 625 measured on the surface without LSP treatment were 410 ± 23.8 HV0.5. After the LSP, the microhardness measured near the surface increased to a value 434 ± 27.5 HV0.5. A 6% increase in microhardness was associated with microstructures generated in the surface region in the irradiated spot during the LSP. Górnikowska et al. [23] and Su et al. [26] found that LSP could significantly improve the microhardness of material surface. In this paper, the main reason why LSP only slightly improved the surface’s microhardness was that the laser energy density was low, the number of shock processing was just three, and the overlap was just 20%. Lv et al. [32] investigated the effects of overlap rate and energy density on mechanical properties of Inconel 718 by multiple small-spot-LSP impacts. It was concluded that the microhardness on LSP-treated surface increased with the rise of the number of impacts and the overlap rate.

### 3.4. Microstructure Characterization of the Fracture

As illustrated in Figure 10, the fracture appearance contains three areas: crack initiation zone, crack propagation zone and ultimate fracture zone.

#### 3.4.1. Crack Initiation Zone

As shown in Figure 11, the fracturing of a specimen with LSP consists of only one crack initiation in general, but for specimens without LSP, there is more than one crack initiation. The plastic deformation of the surface of the material was aggravated by the laser shock wave. LSP made the surface structure more compact by the compressive stress, which increased the tensile resistance of the surface and weakened the structural difference on the surface of the specimen; thus, it was not easy to form a serious stress concentration. Therefore, the specimen with LSP had fewer crack initiations. Moreover, fatigue striations could be observed in the crack initiation zone for all the specimens, as demonstrated in Figure 12a,b,d–f. In addition, the specimen with LSP at 0.4% strain amplitude shows obvious micro cleavage near crack initiation and then exhibits obvious micro fatigue striations away from the crack initiation, as presented in Figure 12c. The crack propagation morphology is closely related to the crack propagation rate. With the increase of the crack propagation rate, the crack propagation morphology went through three stages—microcleavage, striation and microvoid coalescence, discovered by Sullivan et al. [33]. Therefore, Figure 12c reveals that the LSP specimen with 0.4% strain amplitude showed the lowest crack propagation rate. Comparing all the specimens longitudinally and horizontally, the striation space increased with the increase of strain amplitude, and at an identical strain amplitude the fatigue striation space of specimens with LSP was smaller than that of the specimen without LSP. The influence of LSP and strain amplitude on the striation space is discussed in the next section.

#### 3.4.2. Crack Propagation Zone

In the crack propagation zone, all the specimens showed obvious fatigue striations as demonstrated in Figure 13. The fatigue striations were parallel to each other and perpendicular to the crack growth direction. Compared with Figure 13, more secondary cracks parallel to the fatigue striations gradually appeared in this region, and for an identical specimen, the fatigue striations space in the crack propagation zone was larger than that in the crack initiation zone, indicating that the crack growth rate was increasing.

For specimens with different strain amplitudes, the spacing of fatigue striations increased with the increase of strain amplitudes comparing Figure 13a,c,e or comparing Figure 13b,d,f. In addition, with the increase of strain amplitude, the cracks and the intertangle of cracks became more obvious. Moreover, the behavior of the secondary cracks changed obviously with the increase of strain amplitude. In particular, at the lower strain amplitude (0.4%, as shown in Figure 13a,b), only a few secondary cracks appeared, and with the increase of strain amplitude, the density of secondary cracks increased and the entanglement of secondary cracks became more obvious. At the same strain amplitude, the spacing of the fatigue striations of the specimen with LSP was smaller than that of the specimen without LSP, comparing Figure 13a with Figure 13b, Figure 13c with Figure 13d or Figure 13e with Figure 13f. Wang et al. [34] found that the fatigue striations space corresponded to the spreading velocity of crack propagation, i.e., the small fatigue striations space represented low spreading velocity of the crack propagation. As shown in Figure 13, the grain of the specimen with LSP near the surface was significantly refined, and the number of grain boundaries increased after grain refinement, which could arrest the slip of dislocation and crack propagation [35,36]. After the LSP, twins appeared in the subsurface (about 1 μm deep from the surface) of the specimen. The twins generated large angle (60°) grain boundaries, which had an obvious suppression to crack propagation [22,25]. Laser shock wave could produce dislocations in the material surface. During the strain fatigue test, the existing dislocations induced by laser shock would become entangled with the new ones, which would also prevent crack propagation [37,38].

#### 3.4.3. Ultimate Fracture Zone

As shown in Figure 14, all specimens show obvious dimple morphology in the ultimate fracture zone. The dimples of specimens with LSP in the ultimate fracture zone were smaller than those of specimens without LSP, comparing Figure 14a with Figure 14b, Figure 14c with Figure 14d or Figure 14e with Figure 14f. Moreover, microcavities appeared in the ultimate fracture zone of fractures without LSP, as shown in Figure 14b,d,f, but not in fractures with LSP for all strain amplitudes, as shown in Figure 14a,c,e. It was concluded that the surface strengthening of LSP could suppress the formation of dimples in the ultimate fracture zone. Moreover, at about 500 μm depth of the surface affected by LSP, no micropore morphology was observed in the ultimate fracture zone, indicating that LSP made the surface structure dense by residual compressive stress, which inhibited the micropores formation.

In order to further understand the mechanism of dimple formation, EDS spectra at the ultimate fracture zone of the specimen without LSP (0.8%) had been taken for the center of the dimple, as shown in Figure 15a, and margin of the dimple, as shown in Figure 15b. In addition, EDS spectra at the ultimate fracture zone of LSP specimen (0.8%) had been taken to show the effects of LSP on dimple formation, as shown Figure 15c,d. Comparing the EDS spectra at the center of the dimple with the spectra at the margin of the dimple, there was no significant chemical compositional change. The results were different from the research of Pandey [39]. Pandey et al. concluded that the coarsening of carbide particles during loading facilitated the void nucleation that later coalesces, leading to final fracture. Since Inconel 625 only contained about 0.5% carbon, the coarsening of carbide was not obvious or existent. Furthermore, the results of the LSP specimen showed similar chemical composition, which indicated that LSP induced no element precipitation and common chemical reaction, such as oxidation reaction, etc.

The mechanism of fatigue life improved by LSP can be explained from the number of crack initiations, fatigue striations space and size of dimples/micropores. First, one crack initiation leads to one crack propagation path. More crack sources result in larger crack density, causing the lower fatigue life. On the other hand, the width and thickness of the specimen is fixed, which means the distance of the crack propagation is almost identical. In addition, a new fatigue striation occurs in a cycle in the strain fatigue test. Thus, the larger fatigue striations space leads to lower fatigue life. Ultimately, for specimens with LSP, small dimples and the absence of micropores greatly delay the crack propagation, prolonging the fatigue life.

## 4. Conclusions

In the present work, the strain fatigue behavior for Inconel 625 with and without LSP was investigated at different strain amplitudes ranging from 0.4% to 1.2%. The microstructures of specimen surface and fracture were observed by TEM and SEM to clarify the mechanism of fatigue life improvement by LSP. The main conclusions were summarized as followed:(1)LSP could effectively improve the low-cycle fatigue life of Inconel 625.(2)Specimens before and after LSP exhibited similar cyclic strain–stress response behavior as the primary cyclic hardening followed by long-term cyclic softening and approximate maximum cycle stress and cycle number corresponding to the maximum stress.(3)Less values of the number of crack initiations, space of fatigue striations and size of dimples or micropores contributed to the higher fatigue of the specimen after LSP.(4)After LSP, the ultra-fine grains, twins and dislocations appeared at the 1μm depth, which prevented crack initiation, crack propagation and ultimate fracture, increasing the surface’s microhardness and prolonging the fatigue life of Inconel 625.

## Figures and Tables

**Figure 1 materials-15-07269-f001:**
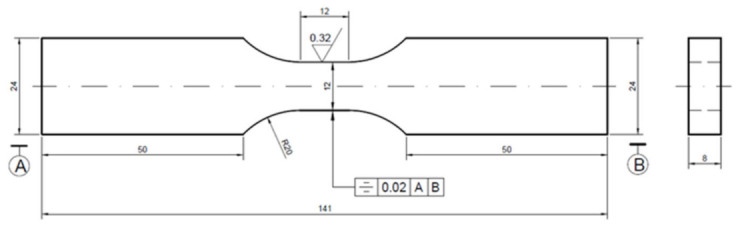
Detailed dimensions of specimens.

**Figure 2 materials-15-07269-f002:**
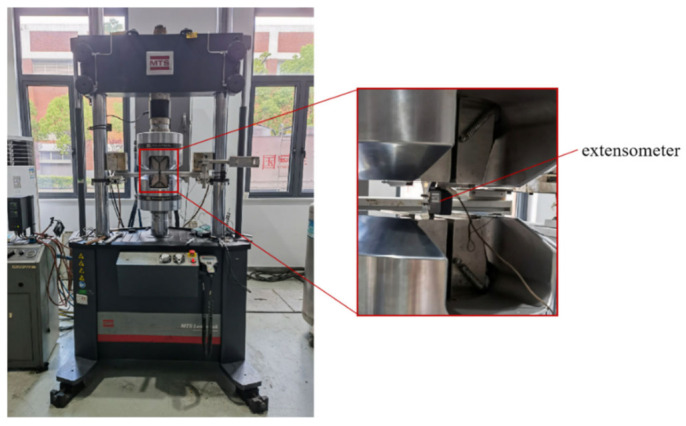
Fatigue machine of strain fatigue tests.

**Figure 3 materials-15-07269-f003:**
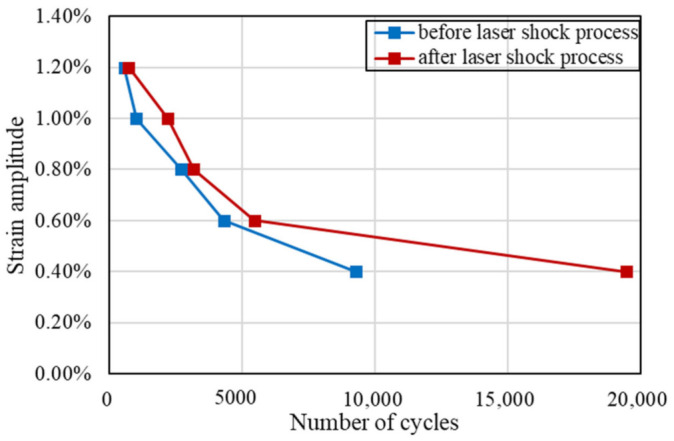
Fatigue life of specimen before and after LSP.

**Figure 4 materials-15-07269-f004:**
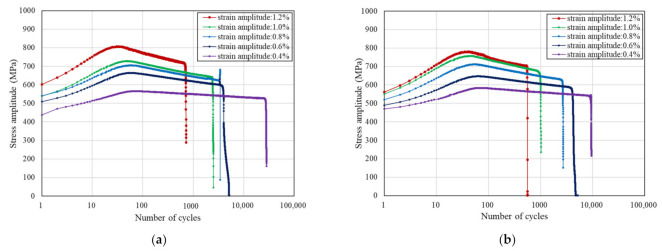
Cyclic strain response to a cycle of controlled deformation for (**a**) Inconel 625 without LSP and (**b**) Inconel 625 with LSP at different strain amplitudes with a constant strain rate of 0.4%/s.

**Figure 5 materials-15-07269-f005:**
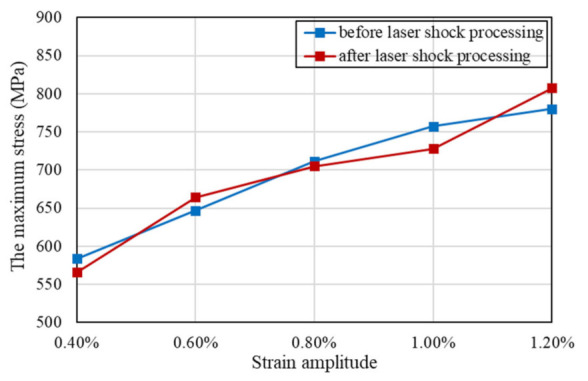
The maximum cyclic stress amplitude of Inconel 625 with and without LSP at different strain amplitudes.

**Figure 6 materials-15-07269-f006:**
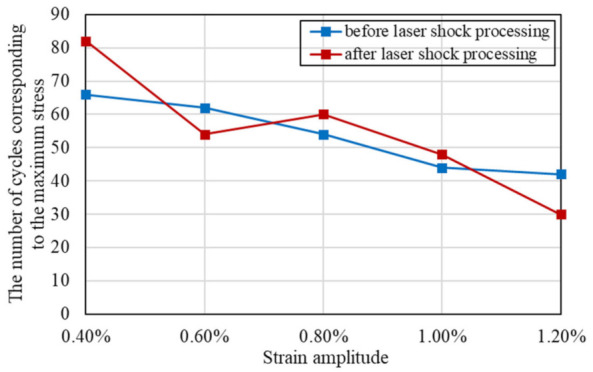
The number of cycles corresponding to the maximum stress at different strain amplitudes.

**Figure 7 materials-15-07269-f007:**
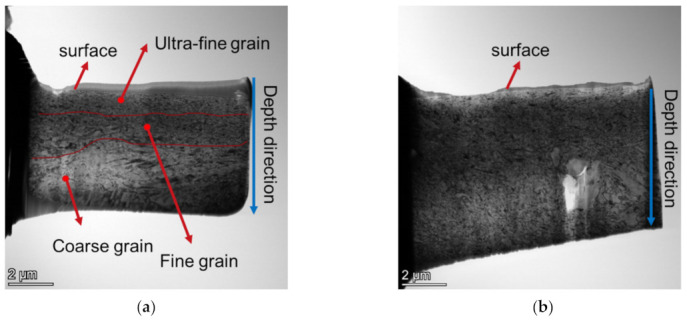
TEM micrographs: (**a**) grain size variation in depth direction of specimen with LSP, (**b**) grain size variation in depth direction of specimen without LSP, (**c**) grain character at different depth of specimen with LSP, and (**d**) grain character at different depth of specimen without LSP.

**Figure 8 materials-15-07269-f008:**
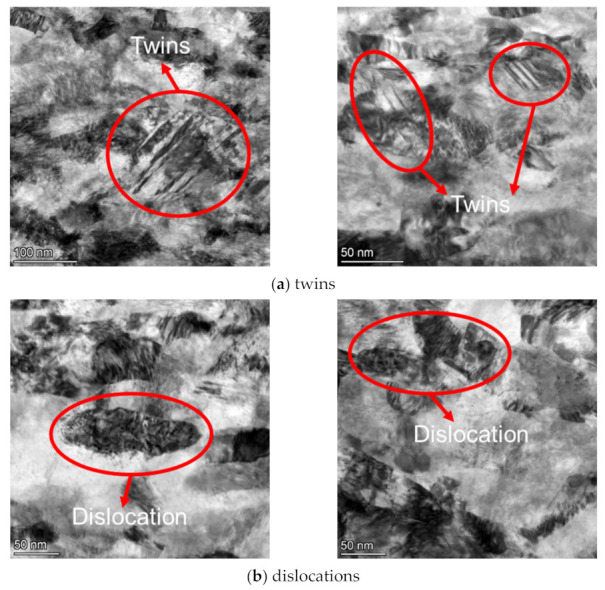
Twins and dislocations at the depth of 1 μm (ultra-fine grain region) of the specimen with LSP.

**Figure 9 materials-15-07269-f009:**
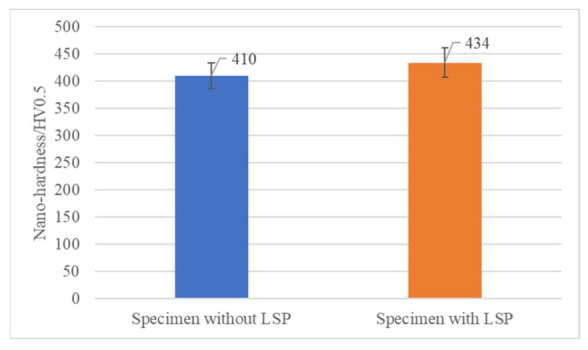
The microhardness of specimens without LSP and specimens with LSP.

**Figure 10 materials-15-07269-f010:**
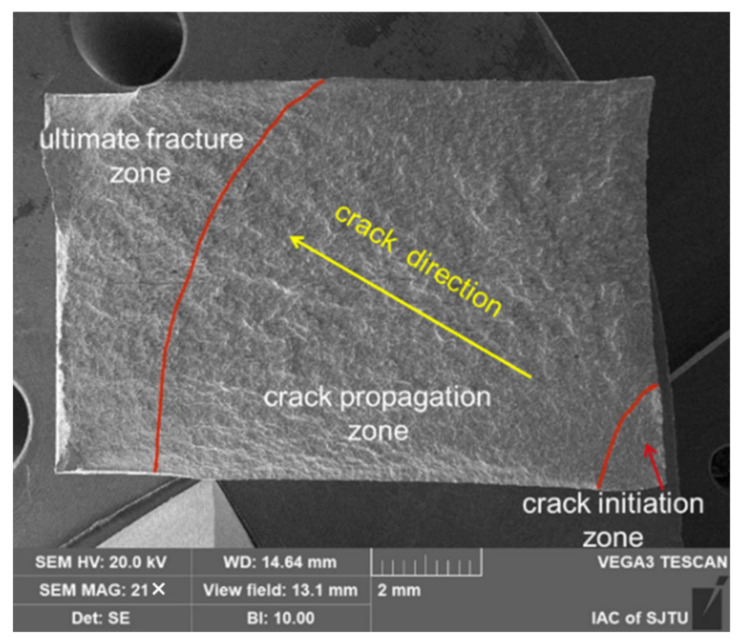
The appearance of fracture for specimens.

**Figure 11 materials-15-07269-f011:**
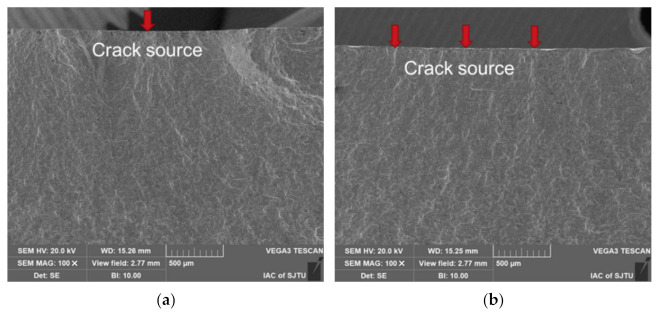
SEM micrographs revealing crack sources for specimens (**a**) with LSP and (**b**) without LSP.

**Figure 12 materials-15-07269-f012:**
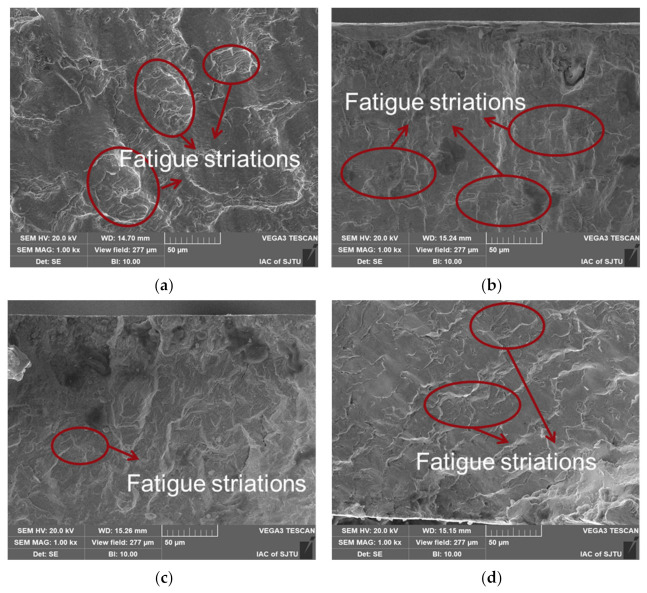
SEM micrographs revealing typical structure in crack initiation zone for specimens with LSP ((**a**) 1.2% strain amplitude, (**c**) 0.8% strain amplitude, (**e**) 0.4% strain amplitude) and specimens without LSP ((**b**) 1.2% strain amplitude, (**d**) 0.8% strain amplitude, (**f**) 0.4% strain amplitude).

**Figure 13 materials-15-07269-f013:**
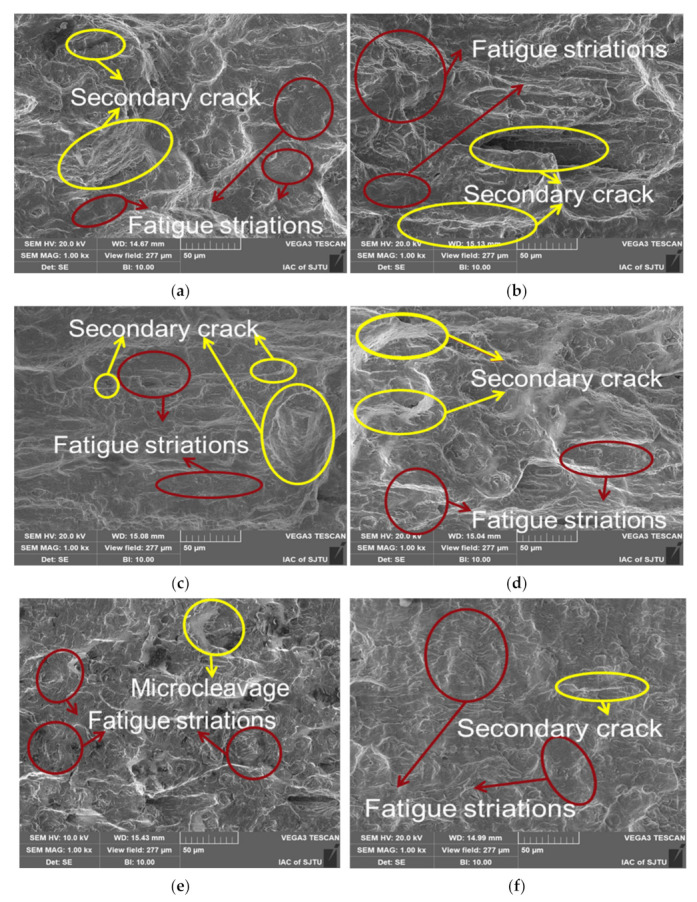
SEM micrographs revealing typical structure in crack propagation zone for specimens with LSP ((**a**) 1.2% strain amplitude, (**c**) 0.8% strain amplitude, (**e**) 0.4% strain amplitude) and specimens without LSP ((**b**) 1.2% strain amplitude, (**d**) 0.8% strain amplitude, (**f**) 0.4% strain amplitude).

**Figure 14 materials-15-07269-f014:**
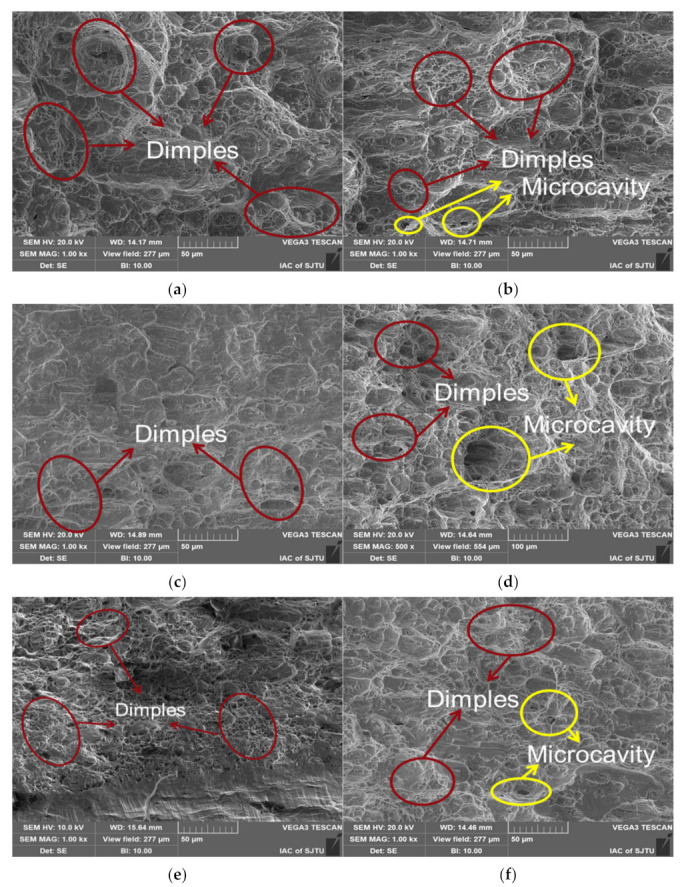
SEM micrographs revealing typical structure in ultimate fracture zone for specimens with LSP ((**a**) 1.2% strain amplitude, (**c**) 0.8% strain amplitude, (**e**) 0.4% strain amplitude) and specimens without LSP ((**b**) 1.2% strain amplitude, (**d**) 0.8% strain amplitude, (**f**) 0.4% strain amplitude).

**Figure 15 materials-15-07269-f015:**
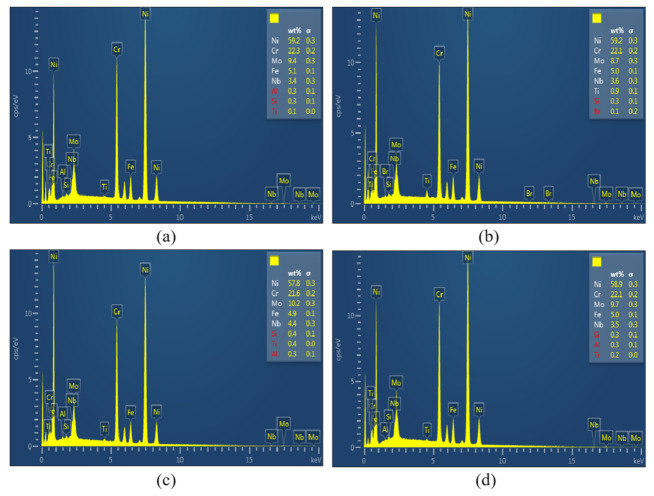
EDS spectra of fracture surface at the ultimate fracture zone for specimens without LSP ((**a**) 0.8% strain amplitude, the center of dimple, (**b**) 0.8% strain amplitude, the margin of dimple) and specimens with LSP ((**c**) 0.8% strain amplitude, the center of dimple, (**d**) 0.8% strain amplitude, the margin of dimple).

**Table 1 materials-15-07269-t001:** Chemical compositions of Inconel 625 (wt.%).

Element	C	Cr	Co	Si	Mo	Mn	Ti	Al	Nb	Fe	Ni
Content %	0.10	23	1.0	0.5	10.0	0.5	0.4	0.4	4.15	5	balance

**Table 2 materials-15-07269-t002:** LSP parameters.

Wavelength	Pulse Width	Repetition Rate	Energy	Laser Spot Size
1064 nm	15 ns	5 Hz	15 J	4 mm

## Data Availability

Data sharing not applicable.

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
