# Peer review of "Investigation of Strain Fatigue Behavior for Inconel 625 with Laser Shock Peening"

_materials, 2022, doi:10.3390/ma15207269_

Round 1
Reviewer 1 Report
Review report: Investigation of strain fatigue behavior for Inconel 625 with laser shock processing
1. Discuss the Novelty of the work.
2. Shorten the length of the conclusion section. Add key conclusion points in the abstract section.
3. Shorten the length of the introduction section and add key published work and try to make a bridge between current and previous published work.
4. How was the composition of the In625 analyzed?
5. Provide complete details of the experimental section along with setup and standard used for specimen preparation.
6. Technical discussion is very week.
7. Fatigue fracture surface needs more discussion. Add the detail regarding it. Add EDS spectra of the fracture surface. Also enhance the quality of the discussion. Refer to following: https://doi.org/10.1016/j.engfailanal.2016.06.012; https://doi.org/10.1016/j.engfailanal.2017.06.044. Add clear information about cleavage area and dimples.
Reviewer 2 Report
The paper is scientifically and technologically on the top, but some aspects need to be clarified or added:
The cyclic tests are mainly in the low-cycle fatigue region and the number of experimental points is very small - only 5 strain levels. Several experimental points are missing to reveal the variance of the results at the different levels of strain amplitudes - only then could a conclusion be drawn on the significance of the effect of LSP on fatigue life (still only in the low-cycle region) on the basis of a statistical comparison.
A comparison with the results of cyclic tests by other authors (e.g. A.Fatemi,...) would be appropriate, at least for material without LSP (the literature sources are limited to a narrow group of authors - this needs to be extended).
Has the chemistry composition in table1 been measured (by what method) ?
"Extensometer" instead of "strain meter" in Fig.2
Chapter 2.2 is better to start with text, not a picture...
Fig.4: "Cyclic strain response to a cycle of controlled deformation for a)..... " instead to author´s text.
Reviewer 3 Report
Authors have investigated the strain fatigue behavoiur
1. Previously lot of work has been carried out.
Rozmus-Górnikowska, M., Kusiński, J. & Cieniek, Ł. Effect of Laser Shock Peening on the Microstructure and Properties of the Inconel 625 Surface Layer. J. of Materi Eng and Perform 29, 1544–1549 (2020). https://doi.org/10.1007/s11665-020-04667-3
2. What is new in the present work
3. Figures 11, 12 and 13 not explained in full. Each and every figure 11a, 11b etc should be explained and presented
4. Fig.12 and Fig.13 not presented inside the text
5. What is the improvement in hardness after LSP
6. Authors are expected to compare the results with other researchers work
Round 2
Reviewer 2 Report
The authors have significantly modified the article in a very positive direction.
I would certainly insist on the clarification that the increase in fatigue properties was demonstrated in the so-called low-cycle region (hence the controlled deformation during cyclic tests).
It should also be clear to the readers whether there are 5 experimental points in Fig. 3 or whether these points are averages of several tests at a given level.
After these clarifications, I recommend publishing the paper.
Reviewer 3 Report
Authors have addressed all the concerns. May be accepted
Author Response
Thanks for the reviewer’s constructive comments. We have double-checked this paper carefully and improved the English language as much as possible.